# Self-Supervised Multi-View Learning via Auto-Encoding 3D Transformations

## Abstract

3D object representation learning is a fundamental challenge in computer vision to draw inferences about the 3D world. Recent advances in deep learning have shown their efficiency in 3D object recognition, among which view-based methods have performed best so far. However, feature learning of multiple views in existing methods is mostly trained in a supervised fashion, which often requires a large amount of data labels with high cost. Hence, it is critical to learn multi-view feature representations in a self-supervised fashion. To this end, we propose a novel self-supervised learning paradigm of Multi-View Transformation Equivariant Representations (MV-TER), exploiting the equivariant transformations of a 3D object and its projected multiple views. Specifically, we perform a 3D transformation on a 3D object, and obtain multiple views before and after transformation via projection. Then, we self-train a representation learning module to capture the intrinsic 3D object representation by decoding 3D transformation parameters from the fused feature representations of multiple views before and after transformation. Experimental results demonstrate that the proposed MV-TER significantly outperforms the state-of-the-art view-based approaches in 3D object classification and retrieval tasks.

## 1 Introduction

3D object representation has become increasingly prominent for a wide range of applications, such as 3D object recognition and retrieval (Maturana & Scherer, 2015; Qi et al., 2016; Brock et al., 2016; Qi et al., 2017a;b; Klokov & Lempitsky, 2017; Su et al., 2015; Feng et al., 2018; Yu et al., 2018; Yang & Wang, 2019). Recent advances in Convolutional Neural Network (CNN) based methods have shown their success in 3D object recognition and retrieval (Su et al., 2015; Feng et al., 2018; Yu et al., 2018; Yang & Wang, 2019). One important family of methods are view-based methods, which project a 3D object into multiple views and learn compact 3D representation by fusing the feature maps of these views for downstream tasks. Feature learning of multiple views in existing approaches are mostly trained in a supervised fashion, hinging on a large amount of data labels that prevents the wide applicability. Hence, self-supervised learning is in demand to alleviate the dependencies on labels by exploring unlabeled data for the training of multi-view feature representations in an unsupervised or (semi-)supervised fashion.

Many attempts have been made to explore self-supervisory signals at various levels of visual structures for representation learning. The self-supervised learning framework requires only unlabeled data in order to formulate a *pretext* learning task (Kolesnikov et al., 2019), where a target objective can be computed without any supervision. These pretext tasks can be summarized into four categories (Jing & Tian, 2019): generation-based (Zhang et al., 2016; Pathak et al., 2016; Srivastava et al., 2015), context-based, free semantic label-based (Faktor & Irani, 2014; Stretcu & Leordeanu, 2015; Ren & Jae Lee, 2018), and cross modal-based (Sayed et al., 2018; Korbar et al., 2018). Among them, context-based pretext tasks include representation learning from image transformations, which is well connected with transformation equivariant representations as they transform equivalently as the transformed images.

Transformation Equivariant Representation learning assumes that representations equivarying to transformations are able to encode the intrinsic structures of data such that the transformations can be reconstructed from the representations before and after transformations (Qi, 2019). Learning

transformation equivariant representations has been advocated in Hinton's seminal work on learning transformation capsules (Hinton et al., 2011). Following this, a variety of approaches have been proposed to learn transformation equivariant representations (Kivinen & Williams, 2011; Sohn & Lee, 2012; Schmidt & Roth, 2012; Skibbe, 2013; Lenc & Vedaldi, 2015; Gens & Domingos, 2014; Dieleman et al., 2015; 2016; Zhang et al., 2019; Qi et al., 2019; Gao et al., 2020; Wang et al., 2020). Nevertheless, these works focus on transformation equivariant representation learning of a single modality, such as 2D images or 3D point clouds.

In this paper, we propose to learn Multi-View Transformation Equivariant Representations (MV-TER) by decoding the 3D transformations from multiple 2D views. This is inspired by the equivariant transformations of a 3D object and its projected multiple 2D views. That is, when we perform 3D transformations on a 3D object, the 2D views projected from the 3D object via fixed viewpoints will transform equivariantly. In contrast to previous works where 2D/3D transformations are decoded from the original single image/point cloud and transformed counterparts, we exploit the equivariant transformations of a 3D object and the projected 2D views. We propose to decode 3D transformations from multiple views of a 3D object before and after transformation, which is taken as self-supervisory regularization to enforce the learning of intrinsic 3D representation. By estimating 3D transformations from the fused feature representations of multiple original views and those of the equivariantly transformed counterparts from the same viewpoints, we enable the accurate learning of 3D object representation even with limited amount of labels.

Specifically, we first perform 3D transformation on a 3D object (*e.g.*, point clouds, meshes), and render the original and transformed 3D objects into multiple 2D views with fixed camera setup. Then, we feed these views into a representation learning module to infer representations of the multiple views before and after transformation respectively. A decoder is set up to predict the applied 3D transformation from the fused representations of multiple views before and after transformation. We formulate multi-view transformation equivariant representation learning as a regularizer along with the loss of a specific task (*e.g.*, classification) to train the entire network end-to-end. Experimental results demonstrate that the proposed method significantly outperforms the state-of-the-art view-based models in 3D object classification and retrieval tasks.

Our main contributions are summarized as follows.

- We propose Multi-View Transformation Equivariant Representations (MV-TER) to learn 3D object representations from multiple 2D views that transform equivariantly with the 3D transformation in a self-supervised fashion.

- We formalize the MV-TER as a self-supervisory regularizer to learn the 3D object representations by decoding 3D transformation from fused features of projected multiple views before and after the 3D transformation of the object.

- Experiments demonstrate the proposed method outperforms the state-of-the-art view-based methods in 3D object classification and retrieval tasks in a self-supervised fashion.

## 2 RELATED WORKS

In this section, we review previous works on transformation equivariant representations and multi-view based neural networks.

### 2.1 TRANSFORMATION EQUIVARIANT REPRESENTATIONS

Many approaches have been proposed to learn equivariant representations, including transforming auto-encoders (Hinton et al., 2011), equivariant Boltzmann machines (Kivinen & Williams, 2011; Sohn & Lee, 2012), equivariant descriptors (Schmidt & Roth, 2012), and equivariant filtering (Skibbe, 2013). Lenc & Vedaldi (2015) prove that the AlexNet (Krizhevsky et al., 2012) trained on ImageNet learns representations that are equivariant to flip, scaling and rotation transformations. Gens & Domingos (2014) propose an approximately equivariant convolutional architecture, which utilizes sparse and high-dimensional feature maps to deal with groups of transformations. Dieleman et al. (2015) show that rotation symmetry can be exploited in convolutional networks for effectively learning an equivariant representation. Dieleman et al. (2016) extend this work to evaluate on other computer vision tasks that have cyclic symmetry. Cohen & Welling (2016) propose group equivariant convolutions that have been developed to equivary to more types of transformations. The idea of

group equivariance has also been introduced to the capsule nets (Lenssen et al., 2018) by ensuring the equivariance of output pose vectors to a group of transformations.

To generalize to generic transformations, Zhang et al. (2019) propose to learn unsupervised feature representations via Auto-Encoding Transformations (AET) by estimating transformations from the learned feature representations of both the original and transformed images. Qi et al. (2019) extend AET by introducing a variational transformation decoder, where the AET model is trained from an information-theoretic perspective by maximizing the lower bound of mutual information. Gao et al. (2020) extend transformation equivariant representations to graph data that are irregularly structured, and formalize graph transformation equivariant representation learning by auto-encoding node-wise transformations in an unsupervised manner. Wang et al. (2020) extend the AET to Generative Adversarial Networks (GANs) for unsupervised image synthesis and representation learning.

## 2.2 MULTI-VIEW LEARNING

Recently, many view-based approaches have been proposed for 3D object learning. These methods project 3D objects (*e.g.*, point clouds, meshes) into multiple views and extract view-wise features receptively via CNNs, and then fuse these features as the descriptor of 3D objects. Su et al. (2015) first propose a multi-view convolutional neural network (MVCNN) to learn a compact descriptor of an object from multiple views, which fuses view-wise features via a max pooling layer. Qi et al. (2016) introduce a new multi-resolution component into MVCNNs, and improve the classification performance. However, max pooling only retains the maximum elements from views, which leads to information loss. In order to address this problem, many subsequent works have been proposed to fuse multiple view-wise features into an informative descriptor for 3D objects. Feng et al. (2018) propose a group-view convolutional neural network (GVCNN) framework, which produces a compact descriptor from multiple views using a grouping strategy. Yu et al. (2018) propose a multi-view harmonized bilinear network (MHBN), which learns 3D object representation by aggregating local convolutional features through the proposed bilinear pooling. To take advantage of the spatial relationship among views, Han et al. (2018) and Han et al. (2019) propose to aggregate the global features of sequential views via attention-based RNN and CNN, respectively. Kanezaki et al. (2018) propose to learn global features by treating pose labels as latent variables which are optimized to self-align in an unsupervised manner. Yang & Wang (2019) propose a relation network to connect corresponding regions from different viewpoints, and reinforce the information of individual view. Jiang et al. (2019) propose a Multi-Loop-View Convolutional Neural Network (MLVCNN) for 3D object retrieval by introducing a novel loop normalization to generate loop-level features. Wei et al. (2020) design a view-based GCN framework to aggregate multi-view features by investigating relations of views.

## 3 MV-TER: THE PROPOSED METHOD

In this section, we first define multi-view equivariant transformation in Section 3.1. Then we formulate the MV-TER model and introduce the MV-TER framework in Section 3.2 and Section 3.4, respectively.

### 3.1 MULTI-VIEW EQUIVARIANT TRANSFORMATION

2D views are projections of a 3D object from various viewpoints, which transform in an equivariant manner as the 3D object transforms. Formally, given a 3D object $\mathbf{M} \in \mathbb{R}^{n \times 3}$ consisting of $n$ points and a 3D transformation distribution $\mathcal{T}$, we sample a transformation $\mathbf{t} \sim \mathcal{T}$ and apply it to $\mathbf{M}$:

$$\widetilde{\mathbf{M}} = \mathbf{t}(\mathbf{M}). \tag{1}$$

We project $\mathbf{M}$ onto 2D views from $m$ viewpoints, denoted as $\mathcal{V} = \{\mathbf{V}_1, ..., \mathbf{V}_m\}$, *i.e.*,

$$\mathbf{V}_i = p_i(\mathbf{M}), \tag{2}$$

where $p_i : \mathbb{R}^3 \mapsto \mathbb{R}^2$ is a projection function for the $i$th view. Subsequent to the transformation on $\mathbf{M}$, the $m$ views transform *equivariantly*, leading to $\widetilde{\mathcal{V}} = \{\widetilde{\mathbf{V}}_1, ..., \widetilde{\mathbf{V}}_m\}$. We have

$$\widetilde{\mathbf{V}}_i = p_i\left(\widetilde{\mathbf{M}}\right) = p_i\left(\mathbf{t}(\mathbf{M})\right) = \mathbf{f}_{i,t}\left(\mathbf{V}_i\right), i = 1, ..., m, \tag{3}$$

where $\mathbf{f}_{i,t}$'s are 2D transformations that are equivariant under the same 3D transformation $\mathbf{t}$. Though $\mathbf{V}_i$ and $\widetilde{\mathbf{V}}_i$ are projected along the same viewpoint $i$ (*i.e.*, the same camera setup), they are projections of the original 3D object and its transformed counterpart, thus demonstrating different perspectives of the same 3D object. Our goal is to learn the representations of 3D objects from their multiple 2D views by estimating the 3D transformation $\mathbf{t}$ as a pretext task from sampled multiple views before and after the transformation, *i.e.*, $\mathcal{V}$ and $\widetilde{\mathcal{V}}$.

## 3.2 THE FORMULATION

Considering the $i$th views $\{\mathbf{V}_i, \widetilde{\mathbf{V}}_i\}$ before and after a transformation $\mathbf{t}$, a function $E(\cdot)$ is *transformation equivariant* if it satisfies

$$E(\widetilde{\mathbf{V}}_i) = E(\mathbf{f}_{i,t}(\mathbf{V}_i)) = \rho(\mathbf{t})E(\mathbf{V}_i), \tag{4}$$

where $\rho(\mathbf{t})$ is a homomorphism of transformation $\mathbf{t}$ in the representation space.

We aim to train a shared representation module $E(\cdot)$ that learns equivariant representations of multiple views. In the setting of self-supervised learning, we formulate MV-TER as a regularizer along with the (semi-)supervised loss of a specific task to train the entire network. Given a neural network with learnable parameters $\Theta$, the network is trained end-to-end by minimizing the weighted sum of two loss functions: 1) the loss of a specific task $\ell_{\text{task}}$ (*e.g.*, a cross-entropy loss in 3D object classification); and 2) the MV-TER loss that is the expectation of estimation error $\ell_{\mathbf{M}}(\mathbf{t}, \hat{\mathbf{t}})$ over each sample $\mathbf{M}$ given a distribution of 3D objects $\mathcal{M}$ and each transformation $\mathbf{t} \sim \mathcal{T}$:

$$\min_{\Theta} \ \ell_{\text{task}} + \lambda \mathop{\mathbb{E}}_{\mathbf{t} \sim \mathcal{T}} \mathop{\mathbb{E}}_{\mathbf{M} \sim \mathcal{M}} \ell_{\mathbf{M}}(\mathbf{t}, \hat{\mathbf{t}}). \tag{5}$$

$\ell_{\mathbf{M}}(\mathbf{t}, \hat{\mathbf{t}})$ is the mean squared error (MSE) between the estimated transformation $\hat{\mathbf{t}}$ and the ground truth $\mathbf{t}$. $\lambda$ is a weighting parameter to strike a balance between the loss of a specific task and the MV-TER loss. Here, the loss $\ell_{\text{task}}$ can be taken over all the data labels (fully-supervised) or partial labels (semi-supervised). In (5), $\hat{\mathbf{t}}$ is decoded as a function of $\mathcal{V}$ and $\widetilde{\mathcal{V}}$ in multiple views as defined in (2) and (3), and we will present two schemes to decode $\hat{\mathbf{t}}$ in the next subsection.

## 3.3 TWO TRANSFORMATION DECODING SCHEMES

**Fusion Scheme.** We propose two schemes to decode the transformation $\mathbf{t}$ in (5) from the feature representations of multiple views $E(\mathbf{V}_i)$ and $E(\widetilde{\mathbf{V}}_i), i = 1, ..., m$. The first scheme is to decode from fused representations of multiple views (before and after transformations). Suppose the neural network extracts features of $\mathbf{V}_i$ and $\widetilde{\mathbf{V}}_i$ from a representation learning module $E(\cdot)$, and estimates the 3D transformation from both features via a transformation decoding module $D(\cdot)$, then we have

$$\hat{\mathbf{t}} = D\left[F\left(E(\mathbf{V}_1), ..., E(\mathbf{V}_m)\right), F\left(E(\widetilde{\mathbf{V}}_1), ..., E(\widetilde{\mathbf{V}}_m)\right)\right], \tag{6}$$

where $F(\cdot)$ is a function of feature fusion.

**Average Scheme.** In the second decoding scheme, we estimate the transformation $\hat{\mathbf{t}}$ from each view before and after transformation and then take the average of the estimates. The idea is each view captures the projected 3D structures under a transformation. This essentially models a 3D object from different perspectives from which the underlying 3D transformation can be revealed. By averaging the estimated transformations across multiple views, a reasonable estimation of 3D transformation can be made.

This actually pushes the model to learn a good 3D representation from individual 2D views and leads to an estimation $\hat{\mathbf{t}}_i$ from the $i$th view:

$$\hat{\mathbf{t}}_i = D\left(E(\mathbf{V}_i), E(\widetilde{\mathbf{V}}_i)\right), i = 1, ..., m. \tag{7}$$

The final decoded 3D transformation is taken as the expectation of $\hat{\mathbf{t}}_i$'s:

$$\hat{\mathbf{t}} = \frac{1}{m} \sum_{i=1}^{m} \hat{\mathbf{t}}_i. \tag{8}$$

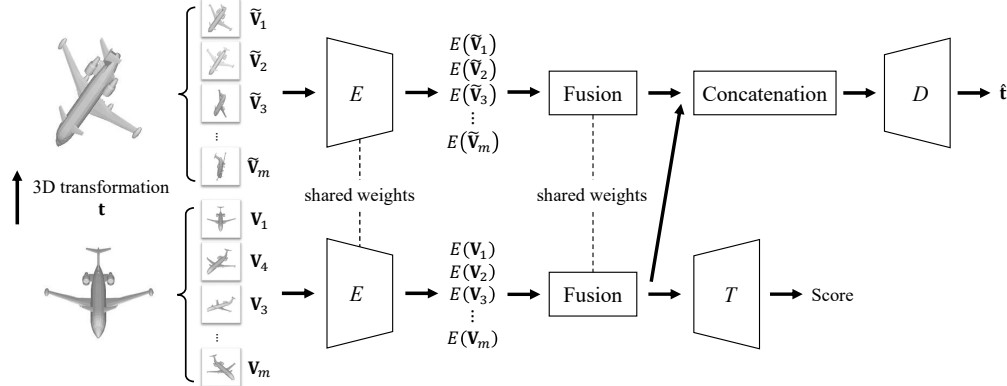

Figure 1: **The architecture of the proposed MV-TER in the fusion decoding scheme.** $E$ and $D$ represent the feature representation module and transformation decoding module respectively, and $T$ is a specific task, *e.g.*, 3D object classification or retrieval.

Hence, we update the parameters $\Theta$ in the representation learning module $E(\cdot)$ and the transformation decoding module $D(\cdot)$ iteratively by backward propagation of the regularized loss in (5).

Interestingly, the second decoding scheme can reach an even better performance than the first scheme in experiments. This should not be surprising since our ultimate goal is to enable multi-view learning by fusing the representations of individual 2D views to reveal the target 3D objects. The second scheme follows this motivation by pushing each view to encode as much information as possible about the 3D transformation, as implied by the multi-view learning.

### 3.4 THE ALGORITHM

Given a 3D object $\mathbf{M}$, we randomly draw a transformation $\mathbf{t} \sim \mathcal{T}$ and apply it to $\mathbf{M}$ to obtain a transformed $\widetilde{\mathbf{M}}$. Then we have $m$ views $\mathcal{V} = \{\mathbf{V}_1, ..., \mathbf{V}_m\}$ by projecting $\mathbf{M}$ to 2D views. Accordingly, the views after the 3D transformation are $\widetilde{\mathcal{V}} = \{\widetilde{\mathbf{V}}_1, ..., \widetilde{\mathbf{V}}_m\}$.

To learn the applied 3D transformation $\mathbf{t}$, we design an end-to-end architecture as illustrated in Figure 1 for the **fusion** decoding scheme, while the architecture of the **average** decoding scheme will be presented in Appendix A. We choose existing CNN models as the representation learning module $E(\cdot)$ (*e.g.*, AlexNet (Krizhevsky et al., 2012), GoogLeNet (Szegedy et al., 2015)), which extract the representation of each view separately. The learned feature representations will be fed into a fusion module and a transformation decoding module $D(\cdot)$ respectively. The fusion module is to fuse the features of multiple views as the overall 3D object representation, *e.g.*, by a view-wise max-pooling layer (Su et al., 2015) or group pooling layer (Feng et al., 2018). The fused feature will serve as the general descriptor of the 3D object for the subsequent downstream learning tasks (*e.g.*, classification and retrieval). The transformation decoding module $D(\cdot)$ is to estimate the 3D transformation parameters from the feature representations of multiple views. Next, we will discuss the representation learning module and transformation decoding module in detail.

#### 3.4.1 REPRESENTATION LEARNING MODULE

The representation learning module $E(\cdot)$ takes the original 2D views $\mathcal{V}$ and their transformed counterparts $\widetilde{\mathcal{V}}$ as the input. $E(\cdot)$ learns feature representations of $\mathcal{V}$ and $\widetilde{\mathcal{V}}$ through a Siamese encoder network with shared weights. Specifically, we employ the feature learning layers of a pre-trained CNN model as the backbone. Then, we obtain the features of each view before and after transformation.

#### 3.4.2 TRANSFORMATION DECODING MODULE

To estimate the 3D transformation $\mathbf{t}$, we concatenate extracted features of multiple views before and after transformation at feature channel, which are then fed into the transformation decoder. The decoder consists of several linear layers to aggregate the representations of multiple views for

Table 1: 3D object classification and retrieval results on ModelNet40 dataset.

| Methods | Training Configuration | | Modality | Classification (Accuracy) | Retrieval (mAP) |
|---|---|---|---|---|---|
| | Pre-train | Fine tune | | | |
| VoxNet (Maturana & Scherer, 2015) | - | ModelNet40 | voxels | 83.0 | - |
| SubvolumeSup (Qi et al., 2016) | - | ModelNet40 | voxels | 89.2 | - |
| Voxception-ResNet (Brock et al., 2016) | - | ModelNet40 | voxels | 91.3 | - |
| PointNet (Qi et al., 2017a) | - | ModelNet40 | points | 89.2 | - |
| PointNet++ (Qi et al., 2017b) | - | ModelNet40 | points | 91.9 | - |
| KD-Networks (Klokov & Lempitsky, 2017) | - | ModelNet40 | points | 91.8 | - |
| MVCNN (Su et al., 2015) | ImageNet1K | ModelNet40 | 12 views | 89.9 | 70.1 |
| MVCNN, metric (Su et al., 2015) | ImageNet1K | ModelNet40 | 12 views | 89.5 | 80.2 |
| MVCNN, multi-resolution (Qi et al., 2016) | ImageNet1K | ModelNet40 | 20 views | 93.8 | - |
| RotationNet (Kanezaki et al., 2018) | ImageNet1K | ModelNet40 | 12 views | 90.7 | - |
| MHBN (Yu et al., 2018) | ImageNet1K | ModelNet40 | 12 views | 93.4 | - |
| SeqViews2SeqLabels (Han et al., 2018) | ImageNet1K | ModelNet40 | 12 views | 93.4 | 89.1 |
| 3D2SeqViews (Han et al., 2019) | ImageNet1K | ModelNet40 | 12 views | 93.4 | 90.8 |
| Relation Network (Yang & Wang, 2019) | ImageNet1K | ModelNet40 | 12 views | 94.3 | 86.7 |
| MLVCNN, Center Loss (Jiang et al., 2019) | ImageNet1K | ModelNet40 | 36 views | 94.2 | 92.8 |
| View-GCN (Wei et al., 2020) | ImageNet1K | ModelNet40 | 12 views | 96.2 | - |
| MVCNN (GoogLeNet) (Feng et al., 2018) | ImageNet1K | ModelNet40 | 12 views | 92.2 | 74.1 |
| MVCNN (GoogLeNet), metric (Feng et al., 2018) | ImageNet1K | ModelNet40 | 12 views | 92.2 | 83.0 |
| **MV-TER (MVCNN), average** | ImageNet1K | ModelNet40 | 12 views | **95.5** | **83.0** |
| **MV-TER (MVCNN), fusion** | ImageNet1K | ModelNet40 | 12 views | **93.1** | **84.9** |
| **MV-TER (MVCNN), average, Center Loss** | ImageNet1K | ModelNet40 | 12 views | **95.1** | **86.6** |
| **MV-TER (MVCNN), fusion, Center Loss** | ImageNet1K | ModelNet40 | 12 views | **93.2** | **89.0** |
| GVCNN (GoogLeNet) (Feng et al., 2018) | ImageNet1K | ModelNet40 | 12 views | 92.6 | 81.3 |
| GVCNN (GoogLeNet), metric (Feng et al., 2018) | ImageNet1K | ModelNet40 | 12 views | 92.6 | 85.7 |
| **MV-TER (GVCNN), average** | ImageNet1K | ModelNet40 | 12 views | **97.0** | **88.8** |
| **MV-TER (GVCNN), fusion** | ImageNet1K | ModelNet40 | 12 views | **96.4** | **88.3** |
| **MV-TER (GVCNN), average, Center Loss** | ImageNet1K | ModelNet40 | 12 views | **95.7** | **91.5** |
| **MV-TER (GVCNN), fusion, Center Loss** | ImageNet1K | ModelNet40 | 12 views | **96.3** | **91.1** |

the prediction of the 3D transformation. As discussed in Section 3.2, we have two strategies for decoding the transformation parameters. We can decode from the fused representations of multiple views before and after transformation as in (6), or from each pair of original and equivariantly transformed views $\{\mathbf{V}_i, \widetilde{\mathbf{V}}_i\}$ to take average for final estimation as in (8). Based on the loss in (5), $\mathbf{t}$ is decoded by minimizing the mean squared error (MSE) between the ground truth and estimated transformation parameters. We will show the estimated 3D transformations of the two decoding schemes in Appendix C.

## 4 EXPERIMENTS

In this section, we evaluate the proposed MV-TER model on two representative downstream tasks: 3D object classification and retrieval.

### 4.1 DATASET

We conduct experiments on the ModelNet40 dataset (Wu et al., 2015). This dataset contains $12,311$ CAD models from $40$ categories. We follow the standard training and testing split settings, *i.e.*, $9,843$ models are used for training and $2,468$ models are for testing. To acquire projected 2D views, we follow the experimental settings of MVCNN (Su et al., 2015) to render multiple views of each 3D object. Here, $12$ virtual cameras are employed to capture views with an interval angle of 30 degree. Next, we employ rotation as our 3D transformations on objects and perform a random rotation with three parameters all in range $[-180°, 180°]$ on the entire 3D object. We also render the views of the transformed 3D object using the same settings as the original 3D object. The rendered multiple views before and after 3D transformations are taken as the input to our method.

### 4.2 3D OBJECT CLASSIFICATION

In this task, we employ the MV-TER as a self-supervisory regularizer to two competitive multi-view based 3D object classification methods: MVCNN (Su et al., 2015) and GVCNN (Feng et al., 2018), which are referred to as **MV-TER (MVCNN)** and **MV-TER (GVCNN)**. Further, we implement with two transformation decoding schemes as discussed in Section 3.3, including the **fusion** scheme and **average** scheme. Then our model has four variants as presented in Table 1.

*Implementation Details:* We deploy GoogLeNet (Szegedy et al., 2015) as our backbone CNN as in GVCNN (Feng et al., 2018). The backbone GoogLeNet is pre-trained on ImageNet1K dataset.

We remove the last linear layer as the Siamese representation learning module to extract features for each view. Subsequent to the representation learning module, we employ one linear layer as the transformation decoding module. The output feature representations of the Siamese network first go through a channel-wise concatenation, which are then fed into the transformation decoding module to estimate the transformation parameters. The entire network is trained via the SGD optimizer with a batch size of $24$. The momentum and weight decay rate are set to $0.9$ and $10^{-4}$, respectively. The initial learning rate is $0.001$, which then decays by a factor of $0.5$ for every $10$ epochs. The weighting parameter $\lambda$ in (5) is set to $1.0$. Also note that, MVCNN in Table 1 has two variants with different backbones: MVCNN (Su et al., 2015) and MVCNN (GoogLeNet) (Feng et al., 2018). MVCNN (Su et al., 2015) uses the VGG-M (Chatfield et al., 2014) as the backbone, while MVCNN (GoogLeNet) implemented in (Feng et al., 2018) employs the GoogLeNet (Szegedy et al., 2015). In addition, RotationNet (Kanezaki et al., 2018) and View-GCN (Wei et al., 2020) are set up with $12$ views taken by the default camera system for fair comparison.

*Experimental Results:* As listed in Table 1, the **MV-TER (MVCNN), average** and **MV-TER (GVCNN), average** achieve classification accuracy of $95.5\%$ and $97.0\%$ respectively, which out-perform the state-of-the-art View-GCN (Wei et al., 2020). Also, the **MV-TER (MVCNN), average** outperforms its baseline MVCNN (GoogLeNet) by $3.3\%$, while **MV-TER (GVCNN), average** out-performs the baseline GVCNN (GoogLeNet) by $4.4\%$, which demonstrates the effectiveness of our proposed MV-TER as a self-supervisory regularizer.

### 4.3 3D Object Retrieval

In this task, we directly employ the fused feature representations of **MV-TER (MVCNN)** and **MV-TER (GVCNN)** as the 3D object descriptor for retrieval. We denote $\mathbf{F}_X$ and $\mathbf{F}_Y$ as the 3D object descriptor of two 3D objects $\mathbf{X}$ and $\mathbf{Y}$ respectively, and use the Euclidean distance between them for retrieval. The distance metric is defined as

$$\text{dist}(\mathbf{X}, \mathbf{Y}) = \|\mathbf{F}_X - \mathbf{F}_Y\|_2. \tag{9}$$

We take the mean average precision (mAP) on retrieval as the evaluation metric, and present the comparison results in the last column of Table 1. For MVCNN and GVCNN, a low-rank Mahalanobis metric learning (Su et al., 2015) is applied to boost the retrieval performance. In comparison, we train our MV-TER model without the low-rank Mahalanobis metric learning, but still achieve better retrieval performance, which validates the superiority of our feature representation learning for 3D objects. Further, we apply Triplet Center Loss (He et al., 2018) to our MV-TER. With Center Loss, our model further achieves an average gain of $3.3\%$ in mAP. As presented in the last column of Table 1, the **MV-TER (GVCNN), average** and **MV-TER (GVCNN), fusion** achieve mAP of $91.5\%$ and $91.1\%$ respectively, which is comparable to MLVCNN with Center Loss (Jiang et al., 2019) while we only take 12 views as input instead of 36 views. We will demonstrate some visual results of 3D object retrieval in Appendix B.

### 4.4 Ablation Studies

#### 4.4.1 On the Number of Views

We quantitatively evaluate the influence of number of views on the classification task. Specifically, we randomly choose $\{8, 12\}$ views from all the views as the input to train **MV-TER (GVCNN), average** respectively, leading to two learned networks. Then, we randomly select $\{2, 4, 8, 12\}$ views from all the testing views to evaluate the classification accuracy of the two networks respectively, as reported in Table 2. We see that we constantly outperform GVCNN with different number of training views and testing views. In particular, when the number of testing views reaches the extreme of two views for multi-view learning, our MV-TER model is still able to achieve the classification accuracy of $91.9\%$ and $91.2\%$, which outperforms GVCNN by a large margin.

#### 4.4.2 On Different Labeling Rates

We adopt six different label rates in the set $\{0.01, 0.02, 0.03, 0.04, 0.05, 0.10\}$ to train four models for comparison: MVCNN (AlexNet), GVCNN (AlexNet), **MV-TER (MVCNN), average** and **MV-TER (GVCNN), average**. When training MVCNN (AlexNet) and GVCNN (AlexNet), we only use a small amount of labeled data to minimize the cross entropy loss for training, and then employ all the test data for evaluation. When training **MV-TER (MVCNN), average** and **MV-TER (GVCNN), average**, we adopt all the data (labeled and unlabeled) to predict the 3D transformations without the

Table 2: Comparison between GVCNN and our **MV-TER (GVCNN), average** with different number of input views for 3D object classification on ModelNet40 dataset. MV-TER is used for brevity.

| Training Views | Testing Views | Accuracy (%) | | Training Views | Testing Views | Accuracy (%) | |
|---|---|---|---|---|---|---|---|
| | | GVCNN | MV-TER | | | GVCNN | MV-TER |
| | 2 | 71.2 | **91.9** | | 2 | 76.8 | **91.2** |
| 8 | 4 | 91.1 | **94.6** | 12 | 4 | 90.3 | **94.3** |
| | 8 | 93.1 | **95.4** | | 8 | 92.1 | **96.1** |
| | 12 | 91.5 | **96.0** | | 12 | 92.6 | **97.0** |

use of labels, and then adopt only labeled data to acquire classification scores. That is, we minimize (5) with a small amount of labels taken for the classification loss $\ell_{\text{task}}$. In all the four models, a pre-trained AlexNet on ImageNet1K dataset is employed as the backbone CNN.

Figure 2 presents the classification accuracy under the six label rates on ModelNet40 dataset. When the label rate is 0.10, we see that the four models achieve comparable results, which benefits from the pre-training of the backbone AlexNet. When the label rate keeps decreasing, the performance of both MVCNN and GVCNN drop quickly, while the MV-TER models are much more robust. Even at the extremely low label rate 0.01, **MV-TER (MVCNN), average** and **MV-TER (GVCNN), average** achieve the classification accuracy of 58.6% and 55.2% respectively, thus demonstrating the robustness of the proposed MV-TER model.

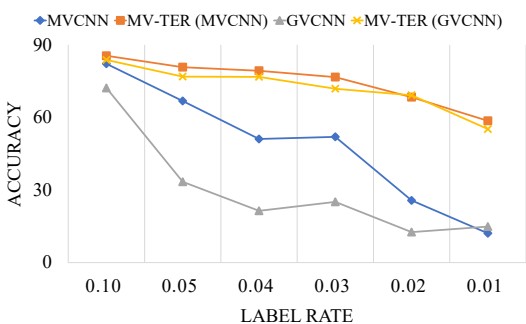

Figure 2: Classification accuracy with different label rates.

### 4.4.3 TRANSFER LEARNING

We further show the generalization performance of the proposed MV-TER under **average** and **fusion** schemes. We take the same network architecture and parameter settings as in Sec. 4.2, except that we set $\lambda = 0.5$ in (5). In particular, we train the feature extractor on the ShapeNetCore55 dataset, and test on the ModelNet40 dataset by a linear SVM classifier using the feature representations of dimension 2048 obtained from the second last fully-connected layer of MV-TER. Table 3 reports the classification comparison of MV-TER and two baseline methods on ModelNet40 dataset under the ShapeNetCore55 pre-training strategies. As we can see, the proposed MV-TER with two decoding schemes improves the average classification accuracy by 2.85% and 2.95% respectively under the ShapeNetCore55 pre-training strategy compared with the two baseline methods MVCNN and GVCNN, thus validating the generalizability.

Table 3: Classification comparison of MV-TER and two baseline methods on ModelNet40 dataset under the ShapeNetCore55 pre-training strategies.

| Method | Accuracy (%) | Method | Accuracy (%) |
|---|---|---|---|
| MVCNN | 85.9 | GVCNN | 87.9 |
| MV-TER, average | **88.7** | MV-TER, average | **91.2** |
| MV-TER, fusion | **89.0** | MV-TER, fusion | **90.3** |

## 5 CONCLUSION

In this paper, we propose a novel self-supervised learning paradigm of Multi-View Transformation Equivariant Representations (MV-TER) via auto-encoding 3D transformations, exploiting the equivariant transformations of a 3D object and its projected multiple views. We perform a 3D transformation on a 3D object, which leads to equivariant transformations in projected multiple views. By decoding the 3D transformation from the fused feature representations of multiple views before and after transformation, the MV-TER enforces the representation learning module to learn intrinsic 3D object representations. Experimental results demonstrate that the proposed MV-TER significantly outperforms the state-of-the-art view-based approaches in 3D object classification and retrieval tasks.

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

# A  ARCHITECTURE OF AVERAGE DECODING SCHEME

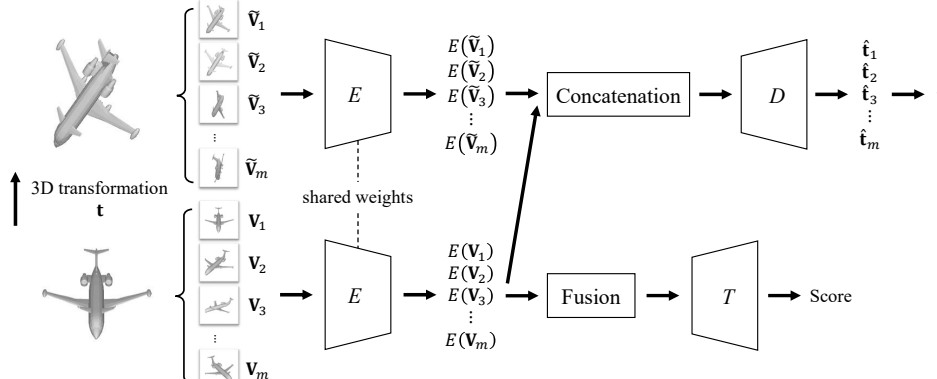

Figure 3: **The architecture of the proposed MV-TER in the average decoding scheme.** $E$ and $D$ represent the feature representation module and transformation decoding module respectively, and $T$ is a specific task, *e.g.*, 3D object classification and retrieval.

We demonstrate the architecture of the proposed MV-TER with the **average** decoding scheme in Figure 3.

# B  VISUALIZATION OF 3D OBJECT RETRIEVAL

Query                                    Top 10 retrieved 3D objects

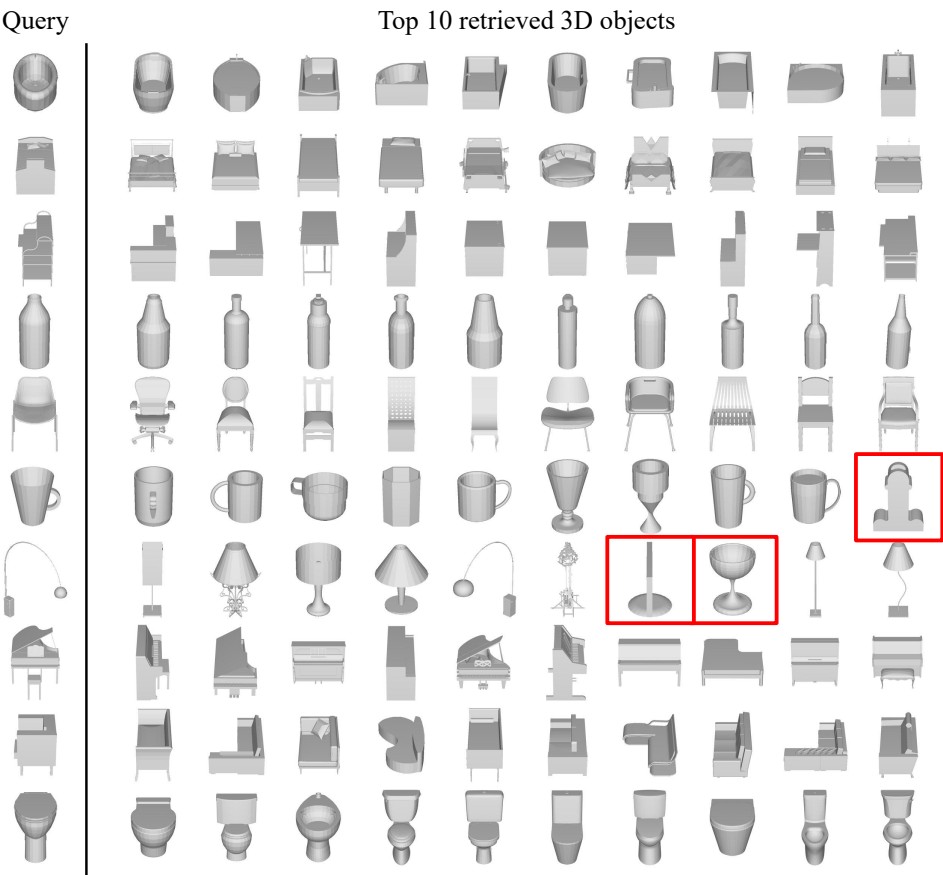

Figure 4: **3D object retrieval examples on ModelNet40 dataset.** Top 10 matches are shown for each query, with mistakes highlighted in red.

We demonstrate some visual results of 3D object retrieval in Figure 4.

# C   EVALUATION OF THE 3D TRANSFORMATION ESTIMATION

Further, to intuitively interpret the estimated 3D transformations from the proposed *fusion* and *average* decoding schemes, we visualize the multiple views projected from 3D objects Car and Bowl with the estimated 3D transformations applied. In Figure 5(a) and Figure 5(b), the first, second and fourth rows demonstrate the projected views from the 3D object with the *same* 3D transformation: the ground truth, the estimation from the *fusion* scheme and the estimation from the *average* scheme. In the third row, each view is the result of each individually estimated 3D transformation $\hat{\mathbf{t}}_i$ as in (7), *i.e.*, view-wise transformations. Note that each column is rendered under the same viewpoint. We see that our MV-TER model estimates more accurate 3D transformations via the *average* scheme, which is consistent with the objective results.

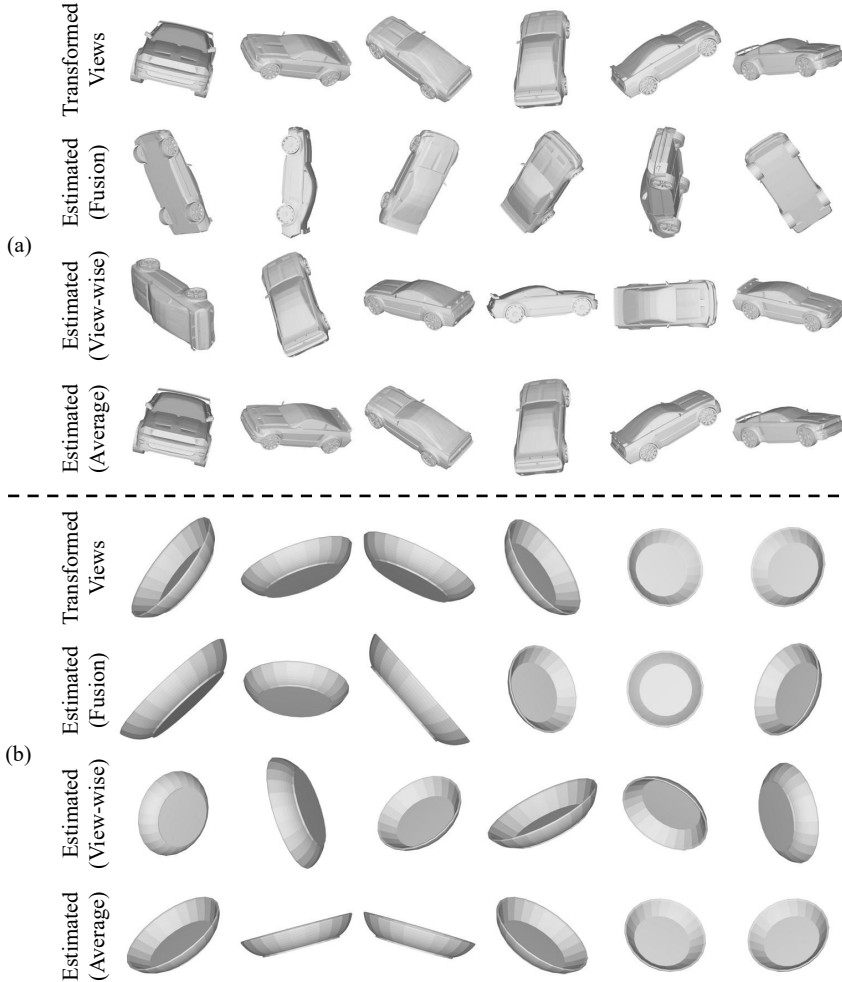

Figure 5: **Illustration of multiple views projected from 3D objects in the same posture: (a) Car and (b) Bowl.** The four rows of (a) and (b) demonstrate multiple views projected from the 3D object with the following 3D transformations applied respectively: 1) the ground-truth 3D transformation; 2) the estimated 3D transformation of the *fusion* decoding scheme; 3) the individually estimated 3D transformations $\hat{\mathbf{t}}_i$'s from each view during the *average* decoding scheme (with $\hat{\mathbf{t}}_i$ applied to the $i$th view); and 4) the finally averaged 3D transformation of the *average* decoding scheme.

Moreover, Figure 6 shows the transformation estimation error on the ModelNet40 dataset under the *average* scheme. The horizontal axis is the index of the training epoch, while the vertical axis refers to the mean squared error. We observe that the MV-TER loss decreases rapidly in the first 40

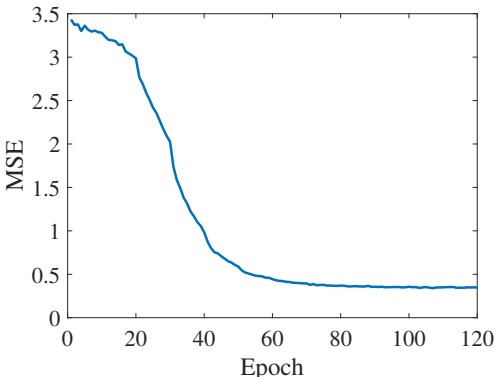

Figure 6: **Transformation estimation error from the *average* scheme.**

epochs. Until the 60th epoch, the mean squared error basically converges to a very small number, thus validating the effectiveness of our model in the transformation estimation.

## D    VISUALIZATION OF FEATURE MAPS

We visualize the feature maps of multiple views projected from 3D objects before and after transformation in Figure 7 for the same category and 8 for different categories. We see that the feature maps of projected multiple views transform equivariantly with the input views. In Figure 7, the feature maps from the same category are similar. In contrast, in Figure 8, although the 3D objects from two different categories are similar, their feature maps are discriminative. This shows the robustness and effectiveness of the learned descriptor.

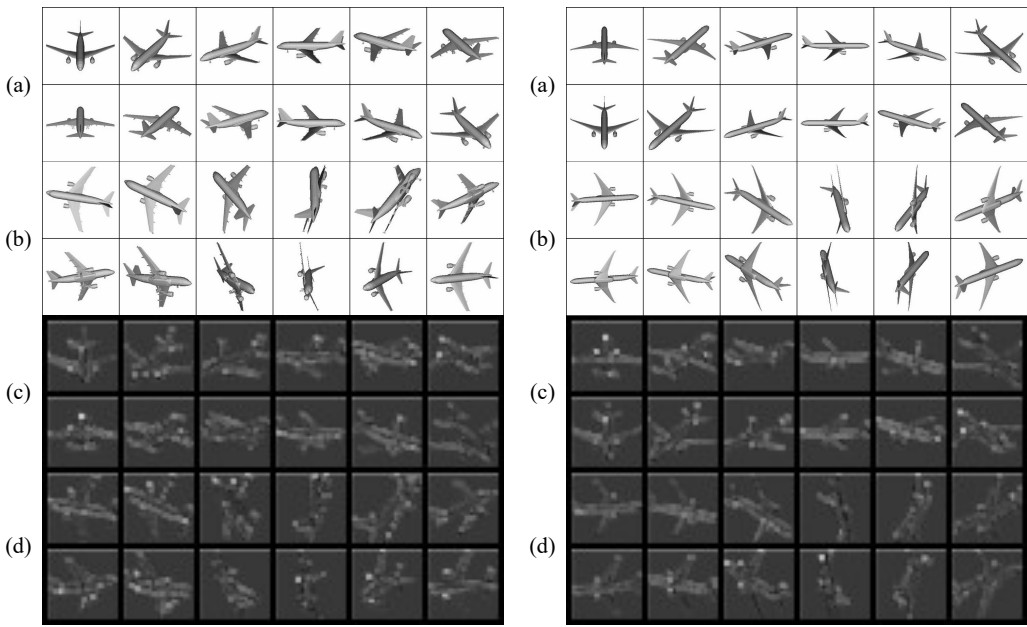

Figure 7: **Illustration of feature maps of multiple views projected from 3D objects before and after transformation in the *same* category Airplane.** (a) and (b) demonstrate multiple views projected from the 3D object before and after transformations, respectively; (c) and (d) show the feature maps of the corresponding views above.

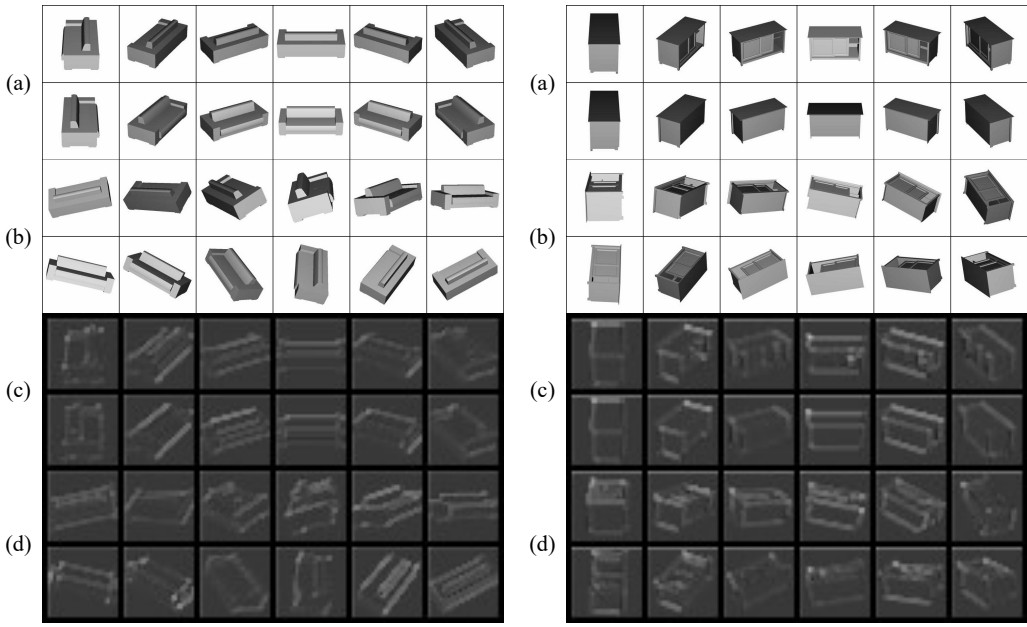

Figure 8: **Illustration of feature maps of multiple views projected from 3D objects before and after transformation in *different* categories Sofa (left) and TV Stand (right).** (a) and (b) demonstrate multiple views projected from the 3D object before and after transformations, respectively; (c) and (d) show the feature maps of the corresponding views above.

