# OpenReview forum: "Self-Supervised Multi-View Learning via Auto-Encoding 3D Transformations"
_ICLR.cc/2021/Conference — Reject_

### Official Review · AnonReviewer4 · 2020-10-28
**The authors propose a simple but effective self-supervised learning technique for multi-view learning.**

**Rating:** 6
**Confidence:** 2

**Review:**

The authors propose a self-supervised learning technique for multi-view learning based on a simple intuition that the transforms of the 2D views of a 3D object will be in an equivariant manner as the 3D object transforms. (Section 3.1). They show its effectiveness by clearly improvements under two frameworks MVCNN and GVCNN.

+ clear presentation
+ simple idea
+ convincing experiments
+ detailed ablation study

Overall, most parts of this paper are satisfactory. They have successfully backed up their claims by applying the idea in two different frameworks and show clear improvements. While my main complaint is that the idea of this work is very simple and even can be called as common sense, and I have encountered this idea in many other papers, though in different fields, such as human pose estimation and 6d object pose estimation. I recommend the authors adding a full discussion of this simple idea based on transformation invariance.

The related works:
3D Human Pose Machines with Self-Supervised Learning, PAMI 2019
Geometry-Driven Self-Supervised Method for 3D Human Pose Estimation, AAAI 2020
Self-Supervised Learning of 3D Human Pose using Multi-view Geometry, CVPR 2019

---

> ### Author Response · Authors · 2020-11-19
> **We thank the reviewer for the valuable comments.**
>
> We thank the reviewer for the valuable comments.
>
> **Q: I recommend the authors adding a full discussion of this simple idea based on transformation invariance.**
>
> **A:** Thank you for the valuable comment.
>
> While our work and the related works provide a bridge between 3D and 2D in general, we particularly exploit transformation EQUIVARIANT representations instead of invariant representations, where the representations of multiple views equivary to projected 3D transformations. The 3D transformation applied to the 3D object links feature representations of multiple views projected from the object. By decoding the applied 3D transformation from the feature representations of multiple 2D views, the model learns 3D information that reveals the intrinsic structure of the 3D object.
>
> We will make a full discussion to the related works and our work.

---

### Official Review · AnonReviewer1 · 2020-10-28
**The manuscript presents a self-supervised learning scheme which improves multi-view object classification and recognition on the modelnet40 dataset. Therefore, I recommend to accept this submission.**

**Rating:** 7
**Confidence:** 4

**Review:**

Summary of the Submission:

This submission proposes a self-supervised learning scheme for 3D object recognition. The basic idea is to predict a 3D transformation from 2D views. The features that facilitate 3D transformation prediction then generalize to other 3D object recognition tasks such as object classification and retrieval. The evaluations on modelnet40 and shapenetcore55 show significant improvement when adding the self-supervised training.

Strengths: Self-supervised formulations are highly beneficial for tasks where no labels are available or where it is expensive to collect labels. The formulation is intuitive, the manuscript is well written and easy to follow.

Update after Rebuttal:

An additional experiment on real data was added which I find a valuable addition to the submission. AR2 points out a similarity with AET which I did not notice in my initial review. I feel this does limit the contribution somewhat. As the rebuttal points out there are differences between the proposed method and AET but the core idea is very similar. In my opinion there is enough difference to still recommend acceptance.

Weakness:

The evaluations are all on synthetic data. It would be great to see how the learned features generalize to real data such as e.g. pascal.

The paper title and the writing is in general talking about 3D transformations but it seems experiments have only been conducted with 3D rotations. Maybe it would be more adequate to change it to 3D rotations everywhere or demonstrate at least one additional type of transformation.

The means squared error as a loss on the 3D rotation is not well motivated. How is the rotation parametrized? Why not using a loss function that penalizes the amount of rotation?

The views are sampled in a regular spacing and not randomly. There is a worry that this induces a bias and recognition with multiple nearby views at test time might be affected as it will not have been present in the training data. But this mode of operation might be a common one in practice if e.g. a video of an object constitutes the multiple views.

This paper seems also related and might be an addition to the related work: Rhodin et al. Unsupervised geometry-aware representation for 3d human pose estimation, ECCV 2018

Explanation of Rating:

This paper shows a self-supervised way to learn features which lead to improved results in classification and recognition on the modelnet40 dataset. The main weakness of the submission is that it was only evaluated on synthetic data. It would be interesting to see if the self-supervised learning helps with generalization to real images.

---

> ### Author Response · Authors · 2020-11-19
> **We thank the reviewer for the valuable comments.**
>
> We thank the reviewer for the valuable comments.
>
> **Q: It would be great to see how the learned features generalize to real data.**
>
> **A:** Thank you for the comment. We evaluate our MV-TER (MVCNN, average) on the real-world multi-view dataset RGBD [*1] in a transfer learning setting. We train the MV-TER on the ShapeNetCore55 dataset, and test on the RGBD dataset by a linear SVM classifier using the feature representations of dimension 2048 obtained from the second last fully-connected layer of MV-TER. We perform 10-fold cross validation to report average accuracies as suggested in [*1]. We compare our method with a baseline method MVCNN under the same evaluation strategy. The MV-TER improves the average classification accuracy by 3.92%, which shows the generalizability of MV-TER to real-world data.
>
> | Method | MV-TER (MVCNN, average) | MVCNN |
> |:-:|:-:|:-:|
> | Accuracy | $\mathbf{68.30 \pm 5.82}$ | $64.38 \pm 5.21$ |
>
> [*1] K. Lai, L. Bo, X. Ren and D. Fox, "A large-scale hierarchical multi-view RGB-D object dataset," 2011 IEEE International Conference on Robotics and Automation, Shanghai, 2011, pp. 1817-1824.
>
> **Q: The paper title and the writing is in general talking about 3D transformations but it seems experiments have only been conducted with 3D rotations.**
>
> **A:** We propose a general framework to learn 3D representations via transformation equivariant representation learning, so any 3D transformation can be applied under our framework. We only use 3D rotations in our paper for simple implementation and clear visualization of experimental results. We will introduce more types of 3D transformations as our future work.
>
> **Q: The means squared error as a loss on the 3D rotation is not well motivated. How is the rotation parametrized? Why not using a loss function that penalizes the amount of rotation?**
>
> **A:** The 3D rotation is parametrized by three Euler angles ranging from $[-\pi,\pi]$, which represent the rotation angles of the x-, y-, z-axis of the 3D object, respectively. Therefore, the proposed loss function actually penalizes the amount of rotation to ensure the equivariance.
>
> **Q: The views are sampled in a regular spacing and not randomly. Multiple nearby views at test time might be affected as it will not have been present in the training data.**
>
> **A:** Though views are projected along fixed viewpoints in the training stage, the proposed model learns the transformation equivariant representations that capture the intrinsic and generalizable representations of multiple views, which is not dependent on how the views are sampled. This is validated by the experimental results in Tab. 2 of the manuscript: we train the MV-TER on 8 views and test on 12 views where some views are not present in the training data. Results show that the MV-TER achieves the classification accuracy of 96.0%, which is comparable to training and testing on 12 views.
>
> **Q: This paper seems also related and might be an addition to the related work in ECCV 2018.**
>
> **A:** Thank you for this comment. We will add this paper to our related work for a more comprehensive review.

---

### Official Review · AnonReviewer2 · 2020-10-29
**Comments for Self-Supervised Multi-View Learning via Auto-Encoding 3D Transformations**

**Rating:** 4
**Confidence:** 5

**Review:**

This paper proposed a self-supervised learning method of 3D shape descriptors for 3D recognition through multi-view 2D image representation learning. To represent the 3D shape, the authors first project the object to a group of 2D project images, which helps apply deep learning due to the image's matrix data format.  The Unsupervised Learning of Transformation Equivariant 2D Representations by Autoencoding Variational Transformations is used for 3D shape descriptor learning, which the authors claimed as "self-supervised" learning. The key idea of transformation equivariant representations is directly borrowed from existing works [1][2]. The method designed is almost the same as [1] except for the encoding network.

[1] Zhang et al., AET vs. AED: Unsupervised Representation Learning by Auto-Encoding Transformations rather than Data, in CVPR 2019. (AET)
[2] Qi et al., AVT: Unsupervised Learning of Transformation Equivariant Representations by Autoencoding Variational Transformations, in ICCV 2019. (AVT)

Besides, there's no proof to verify transformation equivariant representations learning. The authors need to prove the properties for Transformation Equivariant directly for 3D objects. The current presentation of the paper is based on the 2D project image representation for 3D objects. If the authors wish to use Transformation Equivariant  in [1][2] above, the authors might want to consider adding additional 2D to the 3D reconstruction process, and  then consider the order of doing transformation and doing reconstruction etc.)

May some visualization results can better convince the audience. For example, the authors would like to add some visualization results for a 3D shape descriptor for objects from the same category and different categories and show how robust the proposed shape descriptor is.

The proposed approach had experimentally verified its effectiveness in 3D recognition.  However, for a paper being accepted to ICLR, I would like to see more technical novelties/merits beyond directly extending the existing image representation learning approach to 2D projection images of the 3D image. For instance, the authors could propose a method that can be developed based on the "3D Transformation Equivariant" to 3D objects directly instead of its 2D projections.

---Additional comments after rebuttal--
I have carefully reviewed the authors' feedback regarding their comments on how their proposed method differentiates the existing (AVT and AET). Unfortunately, it did not address my concerns about the novel technical contributions in the proposed paper. Obviously, the authors applied the "Transformation Equivariant Representations by Autoencoding Variational Transformations" directly to 2D projections of a 3D object and then fused the deep representation by a shared weight NN (shown in Figure 1). I am not sure why in the rebuttal, the authors claimed, "Their proposed method distinguishes from AET significantly in two aspects". I would urge the authors to check both papers below, and it clearly defines the Transformation Equivariant Representations learning by Autoencoding Variational Transformations, which could be applied for various types of data. [1] Zhang et al., AET vs. AED: Unsupervised Representation Learning by Auto-Encoding Transformations rather than Data, in CVPR 2019. (AET) [2] Qi et al., AVT: Unsupervised Learning of Transformation Equivariant Representations by Autoencoding Variational Transformations, in ICCV 2019. (AVT)

Another concern was critical but not yet addressed neither: The authors could propose a method that can be developed based on the "3D Transformation Equivariant" to 3D objects directly instead of its 2D projections. For this question, I think the authors should prove the Transformation Equivariant Representations directly on a 3D object (point cloud, voxel, 3D mesh) instead of multi-view 2D images. I am not sure why the authors answered that "3D objects are unavailable at the testing stage." The proof of Autoencoding Variational Transformations for 3D data directly should not depend on the availability of 3D data.

---

> ### Author Response · Authors · 2020-11-19
> **We thank the reviewer for the valuable comments.**
>
> We thank the reviewer for the valuable comments.
>
> **Q: The key idea of transformation equivariant representations is directly borrowed from existing works.**
>
> **A:** The proposed method distinguishes from AET significantly in two aspects.
> 1) AET aims to learn equivariant representations of single images by estimating the applied “2D" transformations. In contrast, we focus on representations equivarying to projected 3D transformations onto multiple 2D views by estimating the applied “3D" transformation:
> \begin{equation}
>   E(p_i(\mathbf{t}(\mathbf{M}))) = \rho(\mathbf{t})[E(p_i(\mathbf{M}))],
> \end{equation}
> where $E$ is a representation encoder, $p_i$ is the projection operator for the $i$-th view, $t$ is a 3D transformation, $\mathbf M$ is the 3D object, and $\rho(\cdot)$ is a homomorphism of the 3D transformation $\mathbf t$. Here $p_i \circ \mathbf t$ constructs a composite transformation for the $i$-th view, which leads to various composite transformations for multiple views.
>
> 2) While the projection operator $p_i$ varies for different viewpoints, the acquired representation $E(p_i(\mathbf{t}(\mathbf{M})))$ shares the same 3D transformation $\mathbf t$. That is, the 3D transformation $\mathbf t$ _links_ features of all the views. By decoding the 3D transformation $\mathbf t$ from the feature representations of multiple 2D views, the model learns 3D information that reveals the intrinsic structure of the 3D object.
>
> **Q: The authors could propose a method that can be developed based on the “3D Transformation Equivariant" to 3D objects directly instead of its 2D projections.**
>
> **A:** Thanks for your suggestion. However, we focus on the application scenario of multi-view learning without assuming the availability of 3D objects. That is, 3D objects are unavailable at the testing stage. Hence, we propose representations equivarying to projected 3D transformations onto multiple 2D views by estimating the commonly applied “3D" transformation that links all the views at the training stage.
>
> **Q: Maybe some visualization results can better convince the audience. For example, the authors would like to add some visualization results for a 3D shape descriptor for objects from the same category and different categories and show how robust the proposed shape descriptor is.**
>
> **A:** Thanks for your suggestion. We have visualized the feature maps of multiple views projected from 3D objects before and after transformation from the same category and different categories in Appendix D of the rebuttal revision (marked in blue). We see that the feature maps transform equivariantly with the input views, which validates the transformation equivariance of the learned shape descriptor. Also, feature maps from the same category are similar to each other, while those from different categories are discriminative, which shows the robustness of the learned descriptor.
>
> We hope with our clarifications, the reviewer will see our fundamental contributions.

---

### Official Review · AnonReviewer3 · 2020-10-30
**The paper proposes to use MV-TER loss as a sub-task loss for 3D object classification and retrieval task and improves the performance of STOA methods. However, more detailed analysis and discussion are needed to show the real novelty and value of the loss design to the community.**

**Rating:** 6
**Confidence:** 4

**Review:**

Summary:
This paper proposes a self-supervised learning framework for 3D object classification and retrieval based on multi-view representation, where a sub-task of transformation estimation is adopted as a regularizer. By adding the proposed MV-TER loss, the STOA approaches can gain notable improvement in performance.

Strengths:
-The paper is well written, and the idea of transformation equivariant representation of 3D objects is easy to understand.
-The proposed method further improved the stoa methods on 3D object classification, which is already quite high.

Areas for improvement:
-Firstly, the representation of the transformation that the author used in the paper is not well specified.  I assumed that the Euler angle representation is used in this paper, but I think it is not proper to use the MSE loss in (5) or take the average in (8) for Euler angles.
-Secondly, since it has been suggested by many previous work that the joint-training of multi-task is helpful for the network, it will be appreciated that the authors provide more analysis and discussion on how the MV-TER loss helps the network learn transformation equivariant representation than simply using rotation as data-augmentation or using pose-estimation as sub-task.
-Thirdly, any quantitative results of the transformation estimation? From the images in Appendix B, the results of (average) are incredibly accurate.

---

> ### Author Response · Authors · 2020-11-19
> **We thank the reviewer for the valuable comments.**
>
> We thank the reviewer for the valuable comments.
>
> **Q: The representation of the transformation that the author used in the paper is not well specified. I think it is not proper to use the MSE loss in (5) or take the average in (8) for Euler angles.**
>
> **A:** The 3D rotation transformation is parametrized by three Euler angles ranging from $[-\pi,\pi]$, which represent the rotation angles of the x-, y-, z-axis of the 3D object, respectively. Since we sample discrete rotation transformations, it is reasonable to use the MSE loss for the estimation of 3D transformations and take the average in Eq. (8). We will make this clear in the paper.
>
> **Q: It will be appreciated that the authors provide more analysis and discussion on how the MV-TER loss helps the network learn transformation equivariant representations.**
>
> **A:** Optimizing the MV-TER loss ensures the definition of the transformation equivariance as in Eq. (4), i.e., $E(\widetilde{\mathbf V}_i) = \rho(\mathbf t)E(\mathbf V_i)$, where $E$ is the representation encoder, $\mathbf V_i$ and $\widetilde{\mathbf V}_i$ are views before and after projected 3D transformations, and $\rho(\mathbf t)$ is a homomorphism of the 3D transformation $\mathbf t$. Specifically, we ensure the definition by decoding the 3D transformation parameter $\mathbf t$ from the fused feature representations of multiple views before and after transformation, i.e., $E(\mathbf V_i)$ and $E(\widetilde{\mathbf V}_i)$. The proposed MV-TER loss essentially measures the estimation accuracy of the 3D transformation, thus minimizing the MV-TER loss leads to the learning of transformation equivariant representations. In contrast, the definition of transformation equivariant representations is not guaranteed to be satisfied by simply adopting rotation as data-augmentation or using pose-estimation as a sub-task.
>
>
> **Q: Any quantitative results of the transformation estimation?**
>
> **A:** We present some quantitative results of the transformation estimation (i.e., the MV-TER loss) in the table below. The MV-TER loss reaches 0.59 at the 50th epoch, which is quite small. We also demonstrate detailed quantitative results of transformation estimation from the average scheme in Figure 6 of Appendix C in the rebuttal revision.
>
> | Epoch | 10 | 20 | 30 | 40 | 50 |
> |:-:|:-:|:-:|:-:|:-:|:-:|
> | MV-TER Loss | 3.28 | 2.99 | 2.03 | 0.98 | 0.59 |

---

### Decision · Program_Chairs · 2021-01-07
**Final Decision**

**Decision:**

Reject

**Comment:**

The authors present a method for self-supervised learning of representations of 2D projections of 3D objects. By performing known 3D transformations of an object of interest, a encoder/decoder network is trained to estimate the applied transformation from a series of 2D projections. The proposed method is used as a regularizer and experiments are performed on supervised 3D object classification and retrieval.

After seeing each others’ reviews, one of the main concerns from the reviewers was the relationship between the proposed method and Zhang et al., CVPR 2019 (i.e. AET). The two methods are conceptually very similar, and the consensus from the reviewers is that the authors did not acknowledge the overlap sufficiently and also did not provide a convincing argument as to why they think the approaches are different.

In their rebuttal the authors provided some additional results on real data which is a valuable and welcome addition. However, there were still other concerns that the reviewers had e.g. R2 wanted to know why the model could not be applied directly to 3D shapes instead of 2D projections.

Given the above concerns (specifically the relationship to AET), there is currently not enough support for accepting the paper in its current form. The authors have received detailed feedback and are encouraged to take it onboard when revising the paper in future.